# Comprehensive Prediction and Discriminant Model for Rockburst Intensity Based on Improved Variable Fuzzy Sets Approach

**Hong Wang, Lei Nie, Yan Xu \*, Yan Lv, Yuanyuan He, Chao Du, Tao Zhang and Yuzheng Wang**

Construction Engineering College, Jilin University, Xi Min Zhu Street, Changchun 130026, China
* Correspondence: xuyan8102@jlu.edu.cn

**Abstract:** Rockburst intensity prediction is one of the basic works of underground engineering disaster prevention and mitigation. Considering the dynamic variability and fuzziness in rockburst intensity prediction, variable fuzzy sets (VFS) are selected for evaluation and prediction. Here, there are two problems in the application of traditional VFS: (i) the relative membership degree (RMD) calculation process is complex and time-consuming, and the RMD matrix of all indexes can be only obtained by using the RMD function repeatedly; (ii) unreasonable weights of indicators have great impact on the synthetic relative membership degree (SRMD), so it is difficult to guarantee the correctness of the final prediction result. In view of the above problem, this paper established three simplified feature relationship expressions of RMD based on VFS principle and used the SRMD function to establish a BP neural network model to optimize SRMD. The improved VFS method is more efficient and the prediction results are more stable and reliable than the traditional VFS method. The main advantages are as follows: (1) the improved VFS method has higher computational efficiency; (2) the improved VFS method can verify the correctness of RMD at all times; (3) the improved VFS method has higher prediction accuracy; and (4) the improved VFS method has higher fault tolerance and practicability.

**Keywords:** variable fuzzy sets; rockburst intensity prediction; relative membership degree; synthetic relative membership degree

## 1. Introduction

The frequent occurrence of rockburst disasters has been acknowledged as one of the most serious problems in underground projects all over the world because it directly threatened the safety of underground constructors, equipment, and buildings and even induced mine earthquakes [1]. In recent decades, many reliable measures have been studied to prevent rockburst. For example, Zheng et al. [2] applied borehole pressure relief technology to prevent rockburst in deep-buried mine roadways and achieved good results. Skrzypkowski et al. [3,4] studied a new type of bolt support system, which had good energy absorption and rockburst prevention ability under the condition of tremor. However, in addition to finding reliable prevention and control measures, it is also important to establish an accurate rockburst prediction model in order to resist the occurrence of rockburst disasters [5–8]. Presently, many scholars have studied the mechanism of rockburst from different perspectives and put forward corresponding prediction methods of rockburst intensity, for example, based on single factor strength theory, rigidity theory, energy theory, catastrophe theory, bifurcation theory, instability theory, Russeenes criterion [7], Wang Yuanhan criterion [8] and Lu Jiayou criterion [9], and random forest classification [10], cloud model theory [11], attribute comprehensive evaluation method [12], artificial neural network [13], matter-element extension theory [14], etc. All of the above methods have predicted rockburst from different perspectives and achieved certain prediction results. However,

because rockburst is a very complex nonlinear dynamic phenomenon, it needs many methods to combine and complement each other to accurately predict the intensity of rockburst. Therefore, it is still necessary to introduce new theories and methods to study the occurrence of rockburst and intensity classification prediction.

Based on fuzzy mathematics, Chen [15] has proposed the variable fuzzy sets (VFS) method with relative membership degree (RMD) and synthetic relative membership degree (SRMD) at the core. This method establishes the corresponding VFS model according to the index classification level; then, the RMD function is established by using the VFS model corresponding to each level, and finally the method uses the SRMD function to combine the RMD values and weights of each index to get the final result. Compared with the fuzzy mathematics method, this method considers the dynamic variability and fuzziness of objective things, which improves the reliability of the prediction results [16]. Thus, the VFS method has been widely used in many multi-attribute decision-making problems, such as flood disaster risk assessment [16], surrounding rock stability assessment [17], water resources carrying capacity assessment [18], runoff prediction [19], and so on. Although the indexes and classification criteria are different in different evaluation types, the principle of the VFS method and the characteristics of RMD and SRMD functions are unchanged.

Recently, the research of VFS mainly focuses on the application of the VFS method in different fields and the development of some new methods. For example, Guo et al. [20] applied the VFS model to landslide stability evaluation; Li et al. [21] established a multi-index identification method for flood season stages based on VFS. On the other hand, some researchers also combined VFS with other methods to establish many new evaluation models, such as Ma et al. [22], who combined VFS with fuzzy rough set and applied it to groundwater pollution assessment, and Ren et al. [23], who used evidence theory and VFS to assess flood risk in Chengdu. Fewer studies, however, discuss two core issues of VFS method: (1) the characteristic relationship of RMDs at different classification levels and (2) the influence of index weight on the results of SRMD function calculation.

Chen [24] deduced three laws of dialectics based on VFS and put forward the theory, model, and method of multi-index and multilevel evaluation widely existing in the field of engineering science. However, the characteristics of RMD and SRMD functions in VFS are still not discussed. In the VFS method, the RMD function needs to be run repeatedly to get the final RMD value; this calculation process is very complicated and time-consuming, and the correctness of the calculation results cannot be guaranteed. For SRMD function, a prominent problem is how to reasonably determine the weight of each evaluation index. At present, there are many methods for calculating weights, such as the Delphi method, the AHP method, the efficacy coefficient method, grey relation analysis, and the entropy method. These methods have played an active role in the evaluation and analysis of indicators, but there are still some shortcomings. For example, the Delphi method and AHP are subjective and difficult to implement; the entropy method only depends on the degree of variation of index data, which will lead to deviations in the case of limited information. In addition, the weights calculated by these different methods are quite different and there is no way to verify their rationality, so the accuracy and credibility of the prediction results are reduced. However, the SRMD function has the sigmoid function feature of the BP neural network [15], based on which the BP neural network model can be established to optimize the SRMD so as to effectively reduce the interference of weight on the prediction results and to improve the prediction accuracy of the model method.

Therefore, the primary objectives of this research are (1) to explore the characteristic relationship of RMD in different classifications, of which these features are used to simplify the calculation process of RMD, and (2) to establish a BP neural network model based on SRMD function in order to optimize the SRMD value and to improve the prediction accuracy and practicability of the model. Finally, the improved VFS model and method are applied to the prediction of rockburst cases to verify its feasibility and effectiveness.

## 2. The VFS Method

### 2.1. Principle of VFS

Definition: Let U be the universe of discourse, A be a fuzzy concept in U, and $x$ be a random element, $x \in$ U. The RMDs of $x$ to the concept of attracting (A) and repelling property ($A^c$) are $u_A(x)$ and $u_{A^c}(x)$, respectively. There are the following equations among them:

$$\begin{cases} 0 \leq u_A(x) \leq 1 \\ 0 \leq u_{A^c}(x) \leq 1 \\ u_A(x) + u_{A^c}(x) = 1 \end{cases} \tag{1}$$

Let $D_A(x) = u_A(x) - u_{A^c}(x)$, then $D_A(x) = 2u_A(x) - 1$, where $D_A(x)$ is the relative difference function of $x$ to A. Hence, the RMD can be denoted as follows:

$$u_A(x) = \frac{D_A(x) + 1}{2} \tag{2}$$

### 2.2. RMD Function

In VFS, attraction set and extended set are two important concepts to establish RMD function. Let $X_0 = [a, b]$ be the attracting set of $x$ on the real axis ($0 < D_A(x) \leq 1$) and $X = [c, d]$ be the extended set containing $X_0 (X_0 \in X)$. The M represents equilibrium point $D_A(x) = 1$ in the attracting set $[a, b]$. From the principle of VFS, it is clear that the intervals $[c, a]$ and $[b, d]$ are repelling sets of $x$ ($-1 < D_A(x) \leq 0$) (Figure 1).

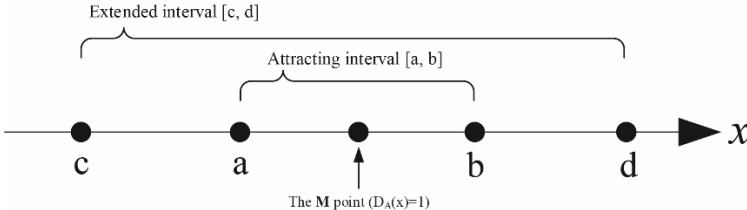

**Figure 1.** The position relation between random point $x$ and intervals $[a, b]$ and $[c, d]$ and point M.

Now, suppose $x$ is a random point among $[c, d]$ (Figure 1); when $x$ is on the left side of M, the relative difference function is established as follows:

$$\begin{cases} D_A(x) = \left(\frac{x-a}{M-a}\right)^\beta & x \in [a, M] \\ D_A(x) = -\left(\frac{x-a}{c-a}\right)^\beta & x \in [c, a] \end{cases} \tag{3}$$

When $x$ is on the right side of M, the relative difference function is established as follows:

$$\begin{cases} D_A(x) = \left(\frac{x-b}{M-b}\right)^\beta & x \in [M, b] \\ D_A(x) = -\left(\frac{x-b}{d-b}\right)^\beta & x \in [b, d] \end{cases} \tag{4}$$

According to Equations (2) and (3), the RMD function is expressed as follows:

$$u_A(x) = 0.5\left[1 + \left(\frac{x-a}{M-a}\right)^\beta\right] \quad x \in [a, M] \tag{5}$$

$$u_A(x) = 0.5\left[1 - \left(\frac{x-a}{c-a}\right)^\beta\right] \quad x \in [c, a] \tag{6}$$

According to Equations (2) and (4), the RMD function is expressed as follows:

$$u_A(x) = 0.5\left[1 + \left(\frac{x-b}{M-b}\right)^{\beta}\right] \quad x \in [M, b] \tag{7}$$

$$u_A(x) = 0.5\left[1 - \left(\frac{x-b}{d-b}\right)^{\beta}\right] \quad x \in [b, d] \tag{8}$$

In Equations (5)–(8), $\beta$ is a nonnegative index and, generally, $\beta = 1$. Equations (5)–(8) meet the following conditions:

$$\begin{cases} u_A(x) = 0.5 \ x = a \ or \ b \\ u_A(x) = 1 \ x = M \\ u_A(x) = 0 \ x \leq c \ or \ x \geq d \end{cases} \tag{9}$$

### 2.3. SRMD Function

In practical applications, the prediction and evaluation objects often have multiple attributes. In order to make the results more realistic, Chen [15] proposed a comprehensive evaluation model to calculate SRMD. The formula is as follows:

$$v(x)_j = \left\{1 + \left[\frac{\sum_{i=1}^{n}\left[w_i\left(1 - u_{ij}(x)\right)\right]^p}{\sum_{i=1}^{n}\left[w_i u_{ij}(x)\right]^p}\right]^{\frac{\alpha}{p}}\right\}^{-1} \quad j = 1, 2, \cdots, m \tag{10}$$

In Equation (10), n and m are the number of indicators and classification grade, respectively; $u_{ij}(x)$ is the RMD of eigenvalue $x$ of the $i$th indicator to level $j$; $\alpha$ and $p$ are the model optimization criterion parameter and the distance parameter, respectively ($\alpha = 1$ for least absolute criteria and $\alpha = 2$ for least squares criterion, $p = 1$ denotes Heming distance and $p = 2$ denotes Euclidean distance); and $w_i$ is the weight of indicator $i$, $\sum_{i=1}^{n} w_i = 1$.

### 2.4. Calculating Steps of VFS Method

Step 1: Establishing parameters (c, a, M, b, d) of VFS model

Assuming an indicator $i$ is divided into m levels and the boundary eigenvalues of each level are $\left([L_{i1}, L_{i2}], [L_{i2}, L_{i3}], \cdots, [L_{im}, L_{i(m+1)}]\right)$ (Figure 2). According to previous literatures [13,22,25], the attracting interval $[a_{ij}, b_{ij}]$, extended interval $[c_{ij}, d_{ij}]$, and point $M_{ij}$ for each level are calculated as follows:

(i)  For the minimum level, the intervals $[a_{i1}, b_{i1}]$ and $[c_{i1}, d_{i1}]$ and point $M_{i1}$ were calculated as follows:

$$\text{Level 1} \begin{cases} c_{i1} = L_{i1} \\ a_{i1} = L_{i1} \\ M_{i1} = L_{i1} \quad i = 1, 2, \cdots, n \\ b_{i1} = L_{i2} \\ d_{i1} = L_{i3} \end{cases} \tag{11}$$

(ii) For level j (j $= 2, 3, \cdots,$ m $- 1$), the intervals $[a_{ij}, b_{ij}]$ and $[c_{ij}, d_{ij}]$ and point $M_{ij}$ were expressed as follows:

$$\text{Level j} \begin{cases} c_{ij} = L_{i(j-1)} \\ a_{ij} = L_{ij} \\ M_{ij} = \left(L_{ij} + L_{i(j+1)}\right)/2 \quad i = 1, 2, \cdots, n \\ b_{ij} = L_{i(j+1)} \\ d_{ij} = L_{i(j+2)} \end{cases} \tag{12}$$

(iii)　For maximum level m, the intervals $[a_{im}, b_{im}]$ and $[c_{im}, d_{im}]$ and point $M_{im}$ were denoted as follows:

$$\text{Level m} \begin{cases} c_{im} = L_{i(m-1)} \\ a_{im} = L_{im} \\ M_{im} = L_{i(m+1)} \quad i = 1, 2, \cdots, n \\ b_{im} = L_{i(m+1)} \\ d_{im} = L_{i(m+1)} \end{cases} \tag{13}$$

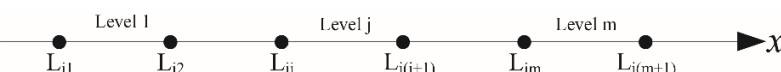

**Figure 2.** The boundary eigenvalues of index *i* corresponding to m levels.

Step 2: Calculating RMD

Substituting the above parameters (c, a, M, b, d) into Equations (5)–(8) repeatedly can obtain RMD functions corresponding to each level. Then, the measured value *x* of index *i* is brought into the RMD function to calculate the RMD value of each level, which is expressed as follows:

$$u_{ij}(x) = [u_{i1}(x), u_{i2}(x), \cdots, u_{im}(x)] \tag{14}$$

Now, assuming that the prediction and assessment object has n indicators, the RMD matrix of n indicators to m levels can be obtained by repeating the above process n times, which can be expressed as follows:

$$u_{ij}(x) = \begin{bmatrix} u_{11}(x) & u_{12}(x) & \cdots & u_{1m}(x) \\ u_{21}(x) & u_{22}(x) & \cdots & u_{2m}(x) \\ \vdots & \vdots & \ddots & \vdots \\ u_{n1}(x) & u_{n2}(x) & \cdots & u_{nm}(x) \end{bmatrix} \tag{15}$$

Step 3: Calculating SRMD

Based on the obtained RMD matrix, the SRMD of the prediction and assessment object to each level can be obtained by introducing $u_{ij}(x)$ into Equation (10):

$$v(x)_j = [v(x)_1, v(x)_2, \cdots, v(x)_m] \tag{16}$$

After calculating the SRMD, the final prediction and assessment level eigenvalue H can be obtained:

$$\begin{cases} H = (1, 2, \cdots, m) \cdot (v(x)_j{}')^T \\ v(x)_j{}' = \dfrac{v(x)_j}{\sum_{j=1}^{m} v(x)_j} \end{cases} \tag{17}$$

The relationship between classification level and eigenvalue H is expressed as follows:

$$\begin{cases} 1 \leq H < 1.5 \ \in \text{Level } 1 \\ j - 0.5 \leq H < j + 0.5 \ \in \text{Level } j \ (j = 1, 2, \cdots, m - 1) \\ m - 0.5 \leq H < m + 0.5 \ \in \text{Level } m \end{cases} \tag{18}$$

In the above calculation process, it can be clearly found that two issues need to be improved when employing the VFS method for multi-index prediction and evaluation. Firstly, how to simplify RMD calculation process? The traditional RMD calculation process is very complicated and time-consuming; a prediction object needs to run n × m times Equations (5)–(8) repeatedly to get the RMD matrix. Although Equation (9) can reduce some, it still has a large computational burden. Secondly, how to

consider the indicator weights of SRMD function so as to obtain high-precision prediction results? In Equation (10), weight assignment is the core issue, which is related to the prediction accuracy of SRMD. However, the weighting is generally carried out in a relatively specific situation in the actual forecasting process. As shown in Table 1, weights are assigned differently based on different occasions, objects, and methods. Weights applicable to one occasion cannot necessarily apply to other occasions, which hinders the promotion and application of prediction methods. Therefore, it is necessary to find a suitable method to optimize SRMD. Only in this way can we effectively guarantee the prediction accuracy of SRMD, ignore the core role of weight, and improve the fault tolerance and application ability of the method.

**Table 1.** The weights of the rockburst prediction factor obtained by different methods.

| No. | Factors | | | Weight Determination Method | Reference |
|---|---|---|---|---|---|
| | $\sigma_\theta/\sigma_c$ | $\sigma_c/\sigma_t$ | $w_{et}$ | | |
| 1 | 0.400 | 0.300 | 0.300 | Fuzzy mathematics | [8] |
| 2 | 0.427 | 0.302 | 0.271 | Back analysis | [26] |
| 3 | 0.365 | 0.313 | 0.322 | Entropy method | [27] |
| 4 | 0.361 | 0.325 | 0.314 | Combination weighting | [28] |
| 5 | 0.162 | 0.675 | 0.162 | Delphi method | [29] |
| 6 | 0.266 | 0.413 | 0.321 | Entropy method | [30] |
| 7 | 0.250 | 0.250 | 0.500 | Rough set theory | [31] |
| 8 | 0.235 | 0.295 | 0.470 | Rough set theory | [32] |

$\sigma_\theta/\sigma_c$: Stress coefficient; $\sigma_c/\sigma_t$: Rock brittleness coefficient; $w_{et}$: Elastic energy index.

## 3. Improved VFS Method

### 3.1. Simplifying the RMD Calculation Process

In the actual calculation process, combining Equations (5)–(8) and (11)–(13), it can be found that, when the measured value $x$ of index $i$ is located in different levels, the RMDs exhibit some general characteristics. Detailed analysis is as follows:

(1) When $x$ is in the minimum level interval, $L_{i1} \le x \le L_{i2}$ (Figure 3a).

When $L_{i1} \le x \le L_{i2}$, only RMD functions of level 1 and level 2 are activated; for other levels, RMDs are equal to 0 because $x$ is not in the $[c, d]$ range. According to Equations (11) and (12), the parameters of the VFS model with levels 1 and 2 can be obtained as follows: $c_{i1} = a_{i1} = M_{i1} = L_{i1}$, $b_{i1} = L_{i2}$, $d_{i1} = L_{i3}$ and $c_{i2} = L_{i1}$, $a_{i2} = L_{i2}$, $M_{i2} = (L_{i2} + L_{i3})/2$, $b_{i2} = L_{i3}$, $d_{i2} = L_{i4}$. Thus, a linear relationship between $x$ and parameters c, a, M, b, and d for levels 1 and 2 can be established as shown in Figure 3b. From Figure 3b, it can be seen that $x$ is in the $[M_{i1}, b_{i1}]$ interval for level 1, and for level 2, $x$ is in the $[c_{i2}, a_{i2}]$ interval. Therefore, the RMD of $x$ for levels 1 and 2 can be calculated by Equations (6) and (7) as follows:

$$\begin{cases} u_{i1}(x) = 0.5\left[1 + \frac{x - b_{i1}}{M_{i1} - b_{i1}}\right] = 0.5\left[1 + \frac{x - L_{i2}}{L_{i1} - L_{i2}}\right] & L_{i1} \le x \le L_{i2} \\ u_{i2}(x) = 0.5\left[1 - \frac{x - a_{i2}}{c_{i2} - a_{i2}}\right] = 0.5\left[1 - \frac{x - L_{i2}}{L_{i1} - L_{i2}}\right] & L_{i1} \le x \le L_{i2} \end{cases} \tag{19}$$

In Equation (19), $0 \le \frac{x - L_{i2}}{L_{i1} - L_{i2}} \le 1$, hence, $0.5 \le u_{i1}(x) \le 1$ and $0 \le u_{i2}(x) \le 0.5$. In addition, it can be easily found that $u_{i1}(x) + u_{i2}(x) = 1$. The above features can also be presented intuitively in Figure 3c. Based on the above analysis, when the measured value $x$ of index $i$ is within level 1, the characteristics of RMD can be summarized as follows:

$$\begin{cases} u_{ij}(x) = [u_{i1}(x), u_{i2}(x), 0, \cdots, 0] \\ u_{i1}(x) + u_{i2}(x) = 1 \\ 0.5 \le u_{i1}(x) \le 1 \\ 0 \le u_{i2}(x) \le 0.5 \end{cases} \tag{20}$$

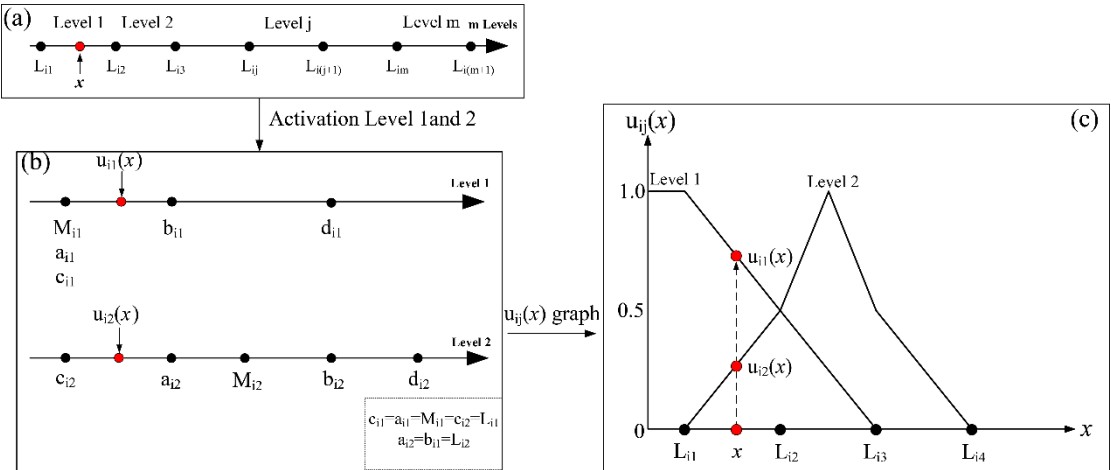

**Figure 3.** (**a**) The linear relationship between measured value $x$ and standard value $L_{ij}$; (**b**) the linear relationship between $x$ and the variable fuzzy set (VFS) model parameters c, a, M, b, and d on activation levels 1 and 2; (**c**) the position of the relative membership degree (RMD) value corresponding to $x$ on the RMD function of activation levels 1 and 2.

(2)　When $x$ is located in the interval of a mid-level j ($j = 2, 3, \cdots, m-1$), $L_{ij} \leq x \leq L_{i(j+1)}$ (Figure 4a).

When $L_{ij} \leq x \leq L_{i(j+1)}$, only RMD functions of level j and adjacent levels j − 1 and j + 1 are activated; for other levels, RMDs are equal to 0 because $x$ is not in the $[c, d]$ range. According to Equation (12), the parameters of the VFS model with levels j, j − 1, and j + 1 can be obtained as follows:

$$\text{Level j} \begin{cases} c_{ij} = L_{i(j-1)} \\ a_{ij} = L_{ij} \\ M_{ij} = (L_{ij} + L_{i(j+1)})/2 & i = 1, 2, \cdots, n \\ b_{ij} = L_{i(j+1)} \\ d_{ij} = L_{i(j+2)} \end{cases} \tag{21}$$

$$\text{Level j − 1} \begin{cases} c_{i(j-1)} = L_{i(j-2)} \\ a_{i(j-1)} = L_{i(j-1)} \\ M_{i(j-1)} = \left(L_{i(j-1)} + L_{ij}\right)/2 & i = 1, 2, \cdots, n \\ b_{i(j-1)} = L_{ij} \\ d_{i(j-1)} = L_{i(j+1)} \end{cases} \tag{22}$$

$$\text{Level j + 1} \begin{cases} c_{i(j+1)} = L_{ij} \\ a_{i(j+1)} = L_{i(j+1)} \\ M_{i(j+1)} = \left(L_{i(j+1)} + L_{i(j+2)}\right)/2 & i = 1, 2, \cdots, n \\ b_{i(j+1)} = L_{i(j+2)} \\ d_{i(j+1)} = L_{i(j+3)} \end{cases} \tag{23}$$

Based on Equations (21)–(23), a linear relationship between $x$ and c, a, M, b, and d for levels j, j − 1, and j + 1 can be established as shown in Figure 4b. From Figure 4b, it can be seen that there are two cases for level j when $x$ is in the $\left[a_{ij}, b_{ij}\right]$ interval: (1) $a_{ij} \leq x \leq M_{ij}$ and (2) $M_{ij} \leq x \leq b_{ij}$. Combined with Equations (5) and (7), the RMD of $x$ for level j can be calculated as follows:

$$\begin{cases} u_{ij}(x) = 0.5\left[1 + \dfrac{x - a_{ij}}{M_{ij} - a_{ij}}\right] = 0.5\left[1 + 2\dfrac{x - L_{ij}}{L_{i(j+1)} - L_{ij}}\right] & L_{ij} \leq x \leq \dfrac{L_{ij} + L_{i(j+1)}}{2} \\ u_{ij}(x) = 0.5\left[1 + \dfrac{x - b_{ij}}{M_{ij} - b_{ij}}\right] = 0.5\left[1 + \dfrac{x - L_{i(j+1)}}{L_{ij} - L_{i(j+1)}}\right] & \dfrac{L_{ij} + L_{i(j+1)}}{2} \leq x \leq L_{i(j+1)} \end{cases} \tag{24}$$

In Equation (24), $0 \leq 2\dfrac{x - L_{ij}}{L_{i(j+1)} - L_{ij}} \leq 1$ and $0 \leq 2\dfrac{x - L_{i(j+1)}}{L_{ij} - L_{i(j+1)}} \leq 1$ such that $0.5 \leq u_{ij}(x) \leq 1$.

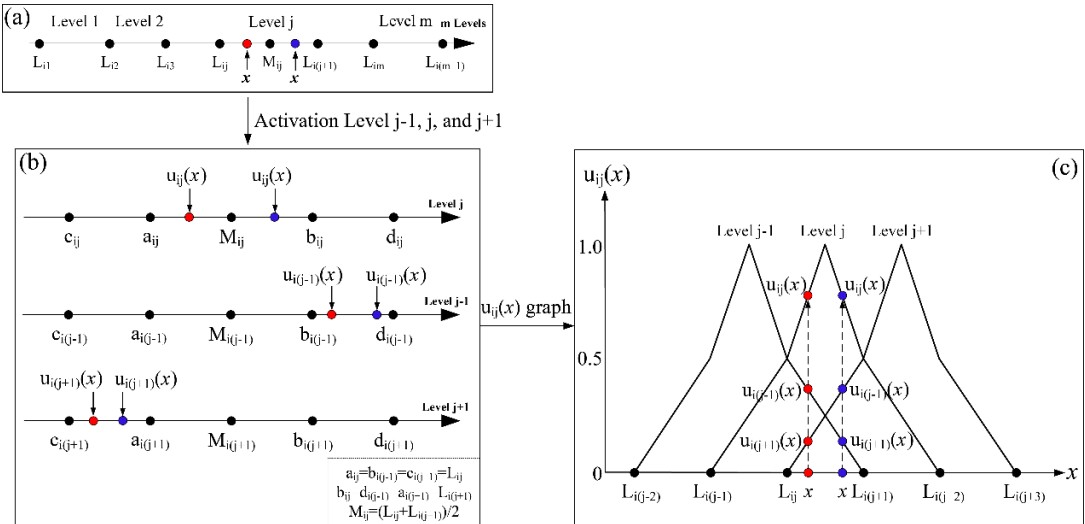

**Figure 4.** (**a**) The linear relationship between measured value $x$ and standard value $L_{ij}$; (**b**) the linear relationship between $x$ and the VFS model parameters c, a, M, b, and d on activation levels j, j − 1, and j + 1; (**c**) the position of the RMD value corresponding to $x$ on the RMD function of activation levels j, j − 1, and j + 1.

Figure 4b also shows that, whether $x$ is in the $\left[a_{ij}, M_{ij}\right]$ or $\left[M_{ij}, b_{ij}\right]$ intervals of level j, in the VFS model of j − 1 and j + 1, $x$ is located in the intervals of $\left[b_{i(j-1)}, d_{i(j-1)}\right]$ and $\left[c_{i(j+1)}, a_{i(j+1)}\right]$, respectively, and RMD can be calculated according to Equations (6) and (8):

$$\begin{cases} u_{i(j-1)}(x) = 0.5\left[1 - \dfrac{x - b_{i(j-1)}}{d_{i(j-1)} - b_{i(j-1)}}\right] = 0.5\left[1 - \dfrac{x - L_{ij}}{L_{i(j+1)} - L_{ij}}\right] L_{ij} \leq x \leq L_{i(j+1)} \\ u_{i(j+1)}(x) = 0.5\left[1 - \dfrac{x - a_{i(j+1)}}{c_{i(j+1)} - a_{i(j+1)}}\right] = 0.5\left[1 - \dfrac{x - L_{i(j+1)}}{L_{ij} - L_{i(j+1)}}\right] L_{ij} \leq x \leq L_{i(j+1)} \end{cases} \tag{25}$$

In Equation (25), $0 \leq \dfrac{x - L_{ij}}{L_{i(j+1)} - L_{ij}} \leq 1$ and $0 \leq \dfrac{x - L_{i(j+1)}}{L_{ij} - L_{i(j+1)}} \leq 1$; hence, $0 \leq u_{i(j-1)}(x) \leq 0.5$ and $0 \leq u_{i(j+1)}(x) \leq 0.5$. In addition, $1 - \dfrac{x - L_{ij}}{L_{i(j+1)} - L_{ij}} + 1 - \dfrac{x - L_{i(j+1)}}{L_{ij} - L_{i(j+1)}} = 2 - 1 = 1$ such that $u_{i(j-1)}(x) + u_{i(j+1)}(x) = 0.5$. The above features can also be clearly seen in Figure 4c, which can be summarized as follows:

$$\begin{cases} u_{ij}(x) = \left[0, \cdots, 0, u_{i(j-1)}(x), u_{ij}(x), u_{i(j+1)}(x), 0, \cdots, 0\right] \\ 0.5 \leq u_{ij}(x) \leq 1 \\ 0 \leq u_{i(j-1)}(x) \leq 0.5 \\ 0 \leq u_{i(j+1)}(x) \leq 0.5 \\ u_{i(j-1)}(x) + u_{i(j+1)}(x) = 0.5 \end{cases} \tag{26}$$

(3) When $x$ is within the maximum level m interval, $L_{im} \leq x \leq L_{i(m+1)}$ (Figure 5a).

When $L_{im} \leq x \leq L_{i(m+1)}$, only RMD functions of level m and level m − 1 are activated; for other levels, RMDs are equal to 0 because $x$ is not in the $[c, d]$ range. According to Equations (12) and (13), the parameters of the VFS model for levels m and m − 1 can be obtained as follows: $c_{im} = L_{i(m-1)}$, $a_{im} = L_{im}$, $M_{im} = b_{im} = d_{im} = L_{i(m+1)}$ and $c_{i(m-1)} = L_{i(m-2)}$, $a_{i(m-1)} = L_{i(m-1)}$, $M_{i(m-1)} = \left(L_{i(m-1)} + L_{im}\right)/2$, $b_{i(m-1)} = L_{im}$, $d_{i(m-1)} = L_{i(m+1)}$. Thus, a linear relationship between $x$ and c, a, M, b, and d for levels m and m − 1 can be established as shown in Figure 5b. Figure 5b shows that

$x$ is in the $[a_{im}, M_{im}]$ interval for level m and that, for level m − 1, x is in the $[b_{i(m-1)}, d_{i(m-1)}]$ interval. Therefore, the RMD of $x$ for levels m and m − 1 can be calculated by Equations (5) and (8) as follows:

$$\begin{cases} u_{im}(x) = 0.5\left[1 + \frac{x-a_{im}}{M_{im}-a_{im}}\right] = 0.5\left[1 + \frac{x-L_{im}}{L_{i(m+1)}-L_{im}}\right] L_{im} \leq x \leq L_{i(m+1)} \\ u_{i(m-1)}(x) = 0.5\left[1 - \frac{x-b_{i(m-1)}}{d_{i(m-1)}-b_{i(m-1)}}\right] = 0.5\left[1 - \frac{x-L_{im}}{L_{i(m+1)}-L_{im}}\right] L_{ij} \leq x \leq L_{i(j+1)} \end{cases} \tag{27}$$

In Equation (27), $0 \leq \frac{x-L_{im}}{L_{i(m+1)}-L_{im}} \leq 1$; hence, $0.5 \leq u_{im}(x) \leq 1$ and $0 \leq u_{i(m-1)}(x) \leq 0.5$. In addition, $1 + \frac{x-L_{im}}{L_{i(m+1)}-L_{im}} + 1 - \frac{x-L_{im}}{L_{i(m+1)}-L_{im}} = 2$ such that $u_{im}(x) + u_{i(m-1)}(x) = 1$. When x is within level m, these features can also be visually found in Figure 5c, summarized as follows:

$$\begin{cases} u_{ij}(x) = \left[0, 0, \cdots, u_{i(m-1)}(x), u_{im}(x)\right] \\ u_{i(m-1)}(x) + u_{im}(x) = 1 \\ 0.5 \leq u_{im}(x) \leq 1 \\ 0 \leq u_{i(m-1)}(x) \leq 0.5 \end{cases} \tag{28}$$

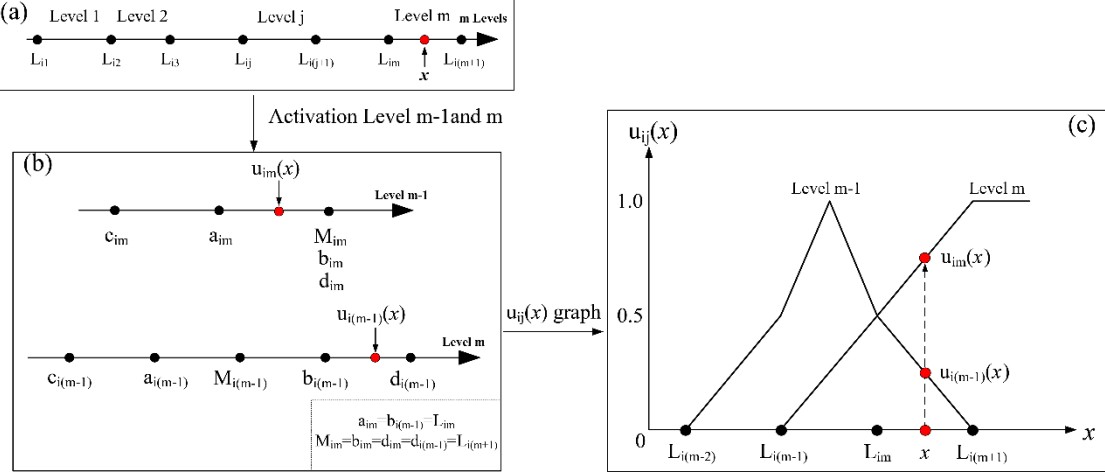

**Figure 5.** (**a**) The linear relationship between measured value $x$ and standard value $L_{ij}$; (**b**) the linear relationship between $x$ and the VFS model parameters c, a, M, b, and d on activation levels m and m − 1; (**c**) the position of the RMD value corresponding to $x$ on the RMD function of activation levels m and m − 1.

In the above analysis, only considering the change of measured value $x$ between the interval $[L_{i1}, L_{i(m+1)}]$, the situation of $x \leq L_{i1}$ or $x \geq L_{i(m+1)}$ is not considered because the cognition is exact and not fuzzy in these two cases; its corresponding RMD value is equal to 1, which is expressed as follows:

$$\begin{cases} u_{ij}(x) = [1, 0, \cdots, 0] \ x \leq L_{i1} \\ u_{ij}(x) = [0, 0, \cdots, 1] \ x \geq L_{i(m+1)} \end{cases} \tag{29}$$

According to the above characteristics (Equations (20), (26), (28), and (29)), combined with Equations (5)–(8), a simplified method of RMD calculation is proposed. The detailed steps are as follows:

(1) The VFS model parameters (c, a, M, b, d) corresponding to each level are established according to the criteria of indicator division, and the position of measured value $x$ in the level interval needs to be identified.

(2) According to the level interval where $x$ is located, the corresponding activation RMD function is determined. If $x$ is in the minimum or maximum levels, it only needs to run the RMD function

once. For level 1, RMD can be calculated by Equation (7), and for the maximum level, RMD can be obtained by Equation (5). If $x$ is in the middle level, only the RMD corresponding to the level itself and one of the adjacent levels need to be calculated for RMD of the level itself, which can be calculated by selecting an appropriate function from Equations (5) and (7), while for the RMD of the adjacent level, one of Equations (6) and (8) can be selected for calculation. The RMD corresponding to other levels can be obtained by feature Equations (20), (26), and (28). In addition, when $x$ is greater than $L_{i(m+1)}$ or less than $L_{i1}$, the RMD function does not need to be run and the RMD is determined directly by Equation (29).

(3) By repeating the above steps, the RMD values for other indicators can be calculated and the final RMD matrix can be obtained.

### 3.2. Optimizing SRMD

In Equation (10), a weight is determined according to the actual situation. If the research target already has a reasonable weight assignment, it is taken as the initialization weight. If it does not exist, equal weight is preferable. Based on this, the obtained RMD is brought into Equation (10) to calculate an initial SRMD. Then, a more accurate result is obtained by optimizing the initial SRMD using the BP network model. In this way, the second issue raised in Section 2.3 can be effectively solved. Many studies have shown that the fusion of BP neural network and fuzzy set theory can give better play to their respective advantages [30,31]. In this paper, we use the sigmoid function characteristics of the SRMD function in Equation (10) to establish a BP neural network to optimize the SRMD. The establishment process mainly includes three parts: (1) creating a network structure; (2) setting the activation function and calculating input and output of each layer; and (3) creating a weight adjustment model. The specific establishment process is as follows:

(1) Creating a network structure

For convenient discussion, a three-layer structure of BP neural network is constructed, as shown in Figure 6. Let the input layer have m nodes, i.e., the initial SRMD corresponds to m levels; the hidden layer have p nodes, i.e., p unit systems; and the output layer have also m nodes corresponding to the optimized SRMD. The error back propagation algorithm of BP network is used to update the connection weights ($w_{ik}$ and $w_{kh}$) between network layers.

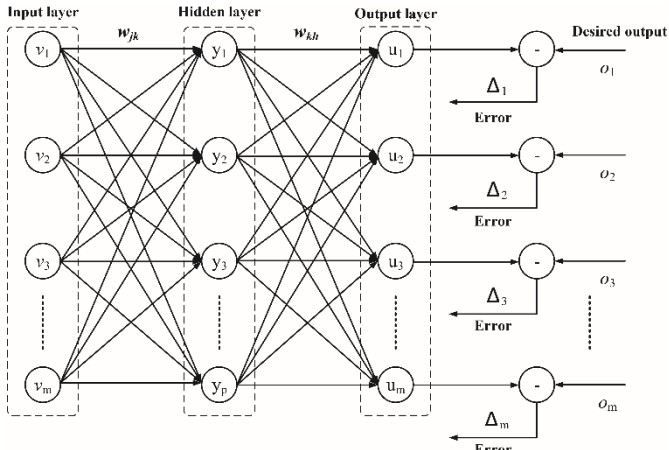

**Figure 6.** BP neural network structure.

(2) Setting the activation function

Sigmoid function is a commonly used activation function of BP networks, which has the characteristics of being s-shaped, asymptotically bounded, completely monotone [32,33]. When $\alpha = 2$,

p $= 1$, let $d_{jg} = \sum_{i=1}^{n}[w_i(1 - u_{ij}(x))]$ and $d_{jb} = \sum_{i=1}^{n} w_i u_{ij}(x)$ such that $d_{jg} = 1 - d_{ib}$. Submit $d_{jb}$ into Equation (10) to obtain the following:

$$v(x)_j = \left[1 + \left(\frac{1 - d_{jb}}{d_{jb}}\right)^2\right]^{-1} \tag{30}$$

According to Equation (30), $v(x)_j$ is a nonlinear function of $d_{jb}$; its first and second derivatives can be expressed as follows:

$$\frac{dv(x)_j}{dd_{jb}} = \frac{2d_{jb}(1 - d_{jb})}{\left((1 - d_{jb})^2 + d_{jb}^2\right)^2} \tag{31}$$

$$\frac{d^2 v(x)_j}{dd_{jb}^2} = \frac{2(1 - 2d_{jb})\left[(1 - d_{jb})^2 + d_{jb}^2 + 4d_{jb}(1 - d_{jb})\right]}{\left[(1 - d_{jb})^2 + d_{jb}^2\right]^2} \tag{32}$$

In Equation (31), $0 \le d_{jb} \le 1$; therefore $\frac{dv(x)_j}{dd_{jb}} > 0$ and then $v(x)_j$ is a monotonically increasing function of $d_{jb}$.

According to Equation (32), when $d_{jb} = 0.5$, $\frac{d^2 v(x)_j}{dd_{jb}^2} = 0$; when $d_{jb} < 0.5$, $\frac{d^2 v(x)_j}{dd_{jb}^2} > 0$; and when $d_{jb} > 0.5$, $\frac{d^2 v(x)_j}{dd_{jb}^2} < 0$. Therefore, the function curve of Equation (30) in the $[0, 0.5]$ interval is concave and in the $[0.5, 1]$ interval is convex, as shown in Figure 7. Based on the above analysis, $d_{jb} = 0.5$ is the only inflection point of Equation (30) in the interval $[0, 1]$, so when $\alpha = 2$ and p $= 1$, the SRMD function (Equation (10)) can be transformed into a sigmoid function.

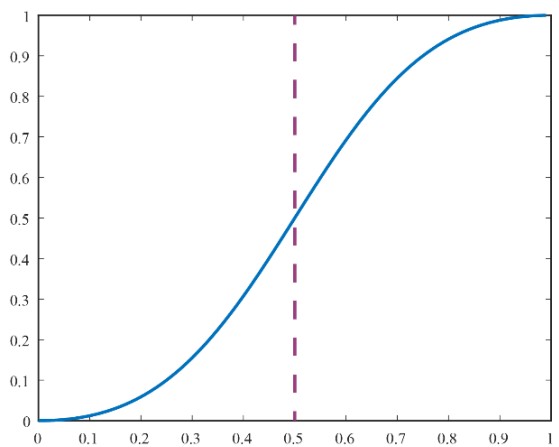

**Figure 7.** The function curve of Equation (30).

(3)   Creating a weight adjustment model

Before creating a weight updating formula, the activation function (Equation (30)) is used to calculate the input and output of hidden and output layers. In the input layer, the input information (SRMD for each level) of node m is transmitted directly to the hidden layer node k, that is, the input and output of the node are equal ($x_j = v(x)_j$). For the hidden layer node k, the input and output are as follows:

$$\begin{cases} I_k = \sum_{j=1}^{m} w_{jk} v(x)_j \\ \\ u_k = \left[1 + \left(\frac{1}{\sum_{j=1}^{m} w_{jk} v(x)_j} - 1\right)^2\right]^{-1} \end{cases} \tag{33}$$

$w_{jk}$ is the connection weight between the input layer and the hidden layer. For the output layer node h, the input and output are as follows:

$$\begin{cases} I_h = \sum_{k=1}^{p} w_{kh}u_k \\ u_h = \left[1 + \left(\dfrac{1}{\sum_{k=1}^{p} w_{kh}u_k} - 1\right)^2\right]^{-1} \end{cases} \tag{34}$$

$w_{kh}$ is the connection weight between the hidden and output layers; $u_h$ is the output value of the output layer of the BP network model, which is the optimized SRMD for level h.

Assuming that the expected output of the evaluation or prediction object is $o_h$, the square error E between the expected output $o_h$ and the actual output $u_h$ is as follows:

$$\mathrm{E} = \frac{1}{2}[o_h - u_h]^2 \tag{35}$$

According to Equation (35), the weight-updating formula between input layer node j and hidden layer node k can be deduced as follows [15]:

$$\begin{cases} \Delta w_{jk} = 2\gamma v(x)_j w_{kh} u_k^2 \left[\dfrac{1 - \sum_{j=1}^{m} w_{jk}v(x)_j}{\left(\sum_{j=1}^{m} w_{jk}v(x)_j\right)^3}\right]\delta_h \\ \delta_h = 2u_h^2 \left[\dfrac{1 - \sum_{k=1}^{p} w_{kh}u_k}{\left(\sum_{k=1}^{p} w_{kh}u_k\right)^3}\right][o_h - u_h] \end{cases} \tag{36}$$

In Equation (36), $\gamma$ is the learning ratio of the BP network. The weight adjustment value $\Delta w_{kh}$ between the hidden layer node k and the output layer node h is as follows:

$$\Delta w_{kh} = 2\gamma u_h^2 u_k \left[\frac{1 - \sum_{k=1}^{p} w_{kh}u_k}{\left(\sum_{k=1}^{p} w_{kh}u_k\right)^3}\right][o_h - u_h] \tag{37}$$

Equations (36) and (37) are the weight adjustment models of the BP network. Applying this model and combining with the iteration algorithm of normal neural network, the connection weight can be determined to minimize the error between the actual output and the expected output.

*3.3. Framework for Assessment and Prediction*

Based on the above analysis, the framework of the assessment and prediction method based on the improved VFS is summarized as follows:

- Determine the impact indicators of the forecasting and evaluation object, determine the classification criteria of impact indicators, and divide them into m levels.
- Determine the VFS model parameters (c, a, M, b, d) of each level according to classification criteria of indicators.
- Calculate the RMDs based on the simplified method proposed in Section 3.1.
- The obtained RMDs are introduced into the comprehensive evaluation model of Equation (10) to initialize SRMD. In Equation (10), the index weight is determined according to the actual situation and equal weight or variable weight is preferable.
- Optimize the initial SRMD according to the BP network model established in Section 3.2.
- The optimized SRMD is introduced into Equation (17) to calculate the eigenvalue H. Then, Equation (18) is used to determine the final prediction and evaluation grade.

The flow chart of this method is shown in Figure 8:

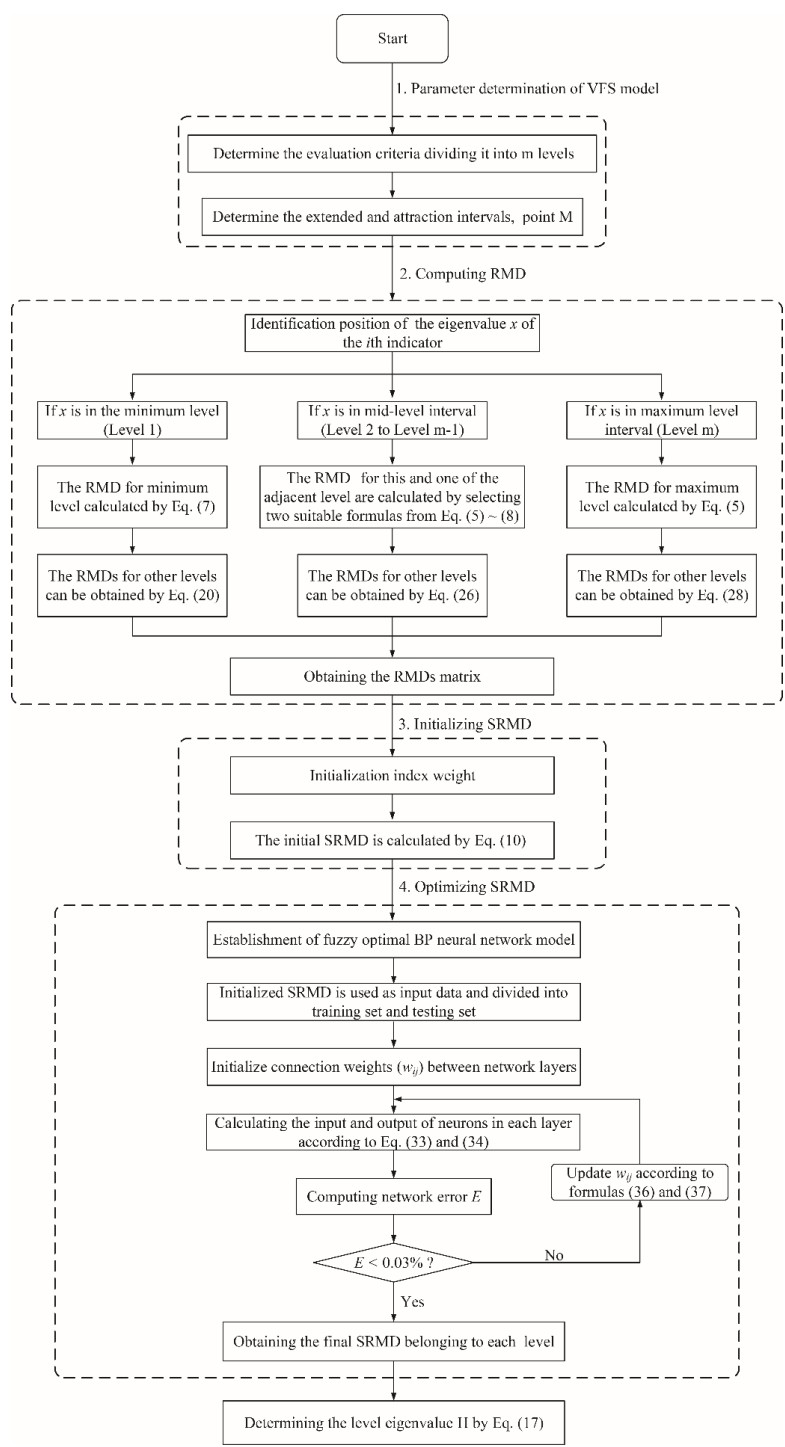

**Figure 8.** Flow chart for improving the VFS assessment and prediction method.

## 4. Results and Discussion

### 4.1. Rockburst Prediction Indicators and Cases

The occurrence mechanism of rockburst is complex and there are many influencing factors. The selection of criteria is the key step in the prediction process. Based on the related studies of rockburst [8,34,35], this paper synthetically considered the factors of in situ stress, lithology, energy, and geological structure and selected the ratio of tangential stress to uniaxial compressive strength $\sigma_\theta/\sigma_c$ (stress coefficient), the ratio of uniaxial compressive strength to tensile strength $\sigma_c/\sigma_t$ (brittleness

coefficient), and elastic energy index $w_{et}$ as indicators of rockburst proneness prediction. From the current research situation, most of the literature has selected similar prediction indicators because they cover both internal and external factors affecting the occurrence of rockburst and are easy to obtain in the laboratory or field [36–38]. Also, it is convenient for comparative analysis between different rockburst engineering cases. According to Wang et al. [8], rockburst prediction criteria for these three indicators are established as shown in Table 2. In addition, in order to verify the correctness and effectiveness of the improved VFS method in rockburst prediction, this paper takes 18 large underground engineering rockburst cases collected by Wang et al. [8] as examples to analyze. The basic data is shown in Table 3.

**Table 2.** The classification criteria of rockburst intensity.

| Levels | Rockburst Indicators | | |
|---|---|---|---|
| | $\sigma_\theta/\sigma_c$ | $\sigma_c/\sigma_t$ | $w_{et}$ |
| None (I) | <0.3 | >40.0 | <2.0 |
| Light (II) | 0.3~0.5 | 26.7~40.0 | 2.0~4.0 |
| Moderate (III) | 0.5~0.7 | 14.5~26.7 | 4.0~6.0 |
| Strong (IV) | >0.7 | <14.5 | >6.0 |

**Table 3.** Basic data of 18 typical rockburst cases around the world.

| No. | Project Name | Main Prediction Indicators | | | Actual Situation |
|---|---|---|---|---|---|
| | | $\sigma_\theta/\sigma_c$ | $\sigma_c/\sigma_t$ | $w_{et}$ | |
| 1 | Diversion tunnel of Tianshengqiao II hydropower station | 0.34 | 23.97 | 6.60 | Moderate |
| 2 | Underground cavern of Longyangxia hydropower station | 0.11 | 31.23 | 7.40 | None |
| 3 | Diversion tunnel of Yuzixi hydropower station | 0.53 | 15.04 | 9.00 | Moderate-strong |
| 4 | Diversion tunnel of Lijiaxia hydropower station | 0.10 | 23.00 | 5.70 | None |
| 5 | Diversion tunnel of Jinping II Hydropower Station | 0.82 | 18.46 | 3.80 | Light-moderate |
| 6 | Underground powerhouse of Sima hydropower station, Norway | 0.27 | 21.69 | 5.00 | Moderate |
| 7 | Sewage tunnel, Norway | 0.42 | 21.69 | 5.00 | Moderate |
| 8 | Diversion tunnel of Vietas hydropower station, Sweden | 0.44 | 26.87 | 5.50 | Light |
| 9 | Guanyue Tunnel, Japan | 0.38 | 28.43 | 5.00 | Moderate-strong |
| 10 | No.2 branch cave of Ertan hydropower station | 0.41 | 29.73 | 7.30 | Light |
| 11 | Underground tunnel of Lubuge hydropower station | 0.23 | 27.78 | 7.80 | None |
| 12 | Diversion tunnel of Taipingyi hydropower station | 0.38 | 17.55 | 9.00 | Moderate |
| 13 | Underground cavern of Pubugou hydropower station | 0.35 | 20.50 | 5.00 | Moderate |
| 14 | Underground powerhouse of Laxiwa hydropower station | 0.32 | 24.11 | 9.30 | Moderate |
| 15 | Heggura Tunnel, Norway | 0.36 | 24.14 | 5.00 | Moderate |
| 16 | Cooling water tunnel of Forsmark nuclear power station, Sweden | 0.39 | 21.67 | 5.00 | Moderate |
| 17 | Mine tunnel of Rasvum chorr, USSR | 0.32 | 21.69 | 5.00 | Moderate |
| 18 | Mine tunnel of Raibl, Italy | 0.77 | 17.50 | 5.50 | Moderate |

## 4.2. The Calculation Process of the Improved VFS Method

(1)　Determining attracting interval, extended interval, and M point

Before calculating the parameters of the VFS model, it should be pointed out that Equations (11)–(13) are available for fixed intervals, while levels I and IV in rockburst prediction criteria are semi-interval forms such as $[-\infty, L_{max}]$ and $[L_{min}, +\infty]$. Here, a pseudo-boundary is obtained using the polynomial regression analysis method, assuming that the trend of increasing or decreasing of boundary eigenvalues is consistent with level L and can be recognized (Figure 9). Based on rockburst classification criteria and the polynomial regression technique, all quantitative boundaries of the levels for all indicators can be obtained and the VFS model parameters $[a_{ij}, b_{ij}]$, $[c_{ij}, d_{ij}]$, and $M_{ij}$ for all level can be determined as follows:

$$\left[a_{ij}, b_{ij}\right] = \begin{bmatrix} [0.1, 0.3] & [0.3, 0.5] & [0.5, 0.7] & [0.7, 0.9] \\ [52.57, 40] & [40, 26.7] & [26.7, 14.5] & [14.5, 1.57] \\ [0.5, 2] & [2, 3.5] & [3.5, 5] & [5, 6.5] \end{bmatrix} \qquad (38)$$

where the first row is the attracting interval of $\sigma_\theta/\sigma_c$ to four levels and the remaining rows represent $\sigma_c/\sigma_t$ and $w_{et}$.

$$\left[M_{ij}\right] = \begin{bmatrix} 0.1 & 0.4 & 0.6 & 0.9 \\ 52.57 & 33.35 & 20.6 & 1.57 \\ 0.5 & 2.75 & 4.25 & 6.5 \end{bmatrix} \tag{39}$$

where the first row is the points M of $\sigma_\theta/\sigma_c$ to four levels and the remaining rows represent $\sigma_c/\sigma_t$ and $w_{et}$.

$$\left[c_{ij}, d_{ij}\right] = \begin{bmatrix} [0.1, 0.5] & [0.1, 0.7] & [0.3, 0.9] & [0.5, 0.9] \\ [52.57, 26.7] & [52.57, 14.5] & [40, 1.57] & [26.7, 1.57] \\ [0.5, 3.5] & [0.5, 5] & [2, 6.5] & [3.5, 6.5] \end{bmatrix} \tag{40}$$

where the first row is the extended interval of $\sigma_\theta/\sigma_c$ to four levels and the remaining rows represent $\sigma_c/\sigma_t$ and $w_{et}$.

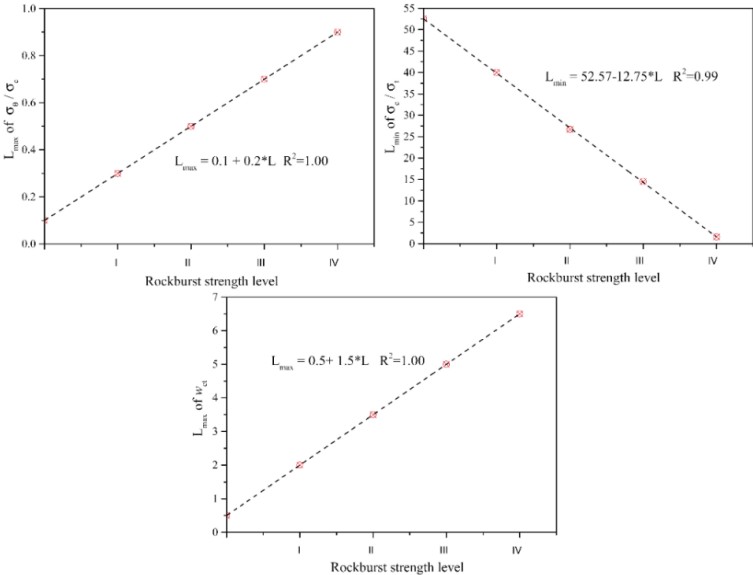

**Figure 9.** Polynomial regression of the quantitative boundary eigenvalue ($L_{min}$ or $L_{max}$) for all indicators.

(2)  RMD matrix of indicators

In this paper, the rockburst case in Tianshengqiao II hydropower station is selected to illustrate the calculation process of RMD. The eigenvalues of $\sigma_\theta/\sigma_c$, $\sigma_c/\sigma_t$, and the $w_{et}$ indicator in Tianshengqiao II hydropower station are 0.34, 23.97, and 6.60, respectively.

The eigenvalue ($x_1 = 0.34$) of $\sigma_\theta/\sigma_c$ is located in the interval of level II. For level II, the VFS model parameters (c, a, M, b, d) are 0.1, 0.3, 0.4, 0.5, and 0.7, respectively. From this, it can be seen that the eigenvalue $x_1$ is between a (0.3) and M (0.4). Taking these parameters into Equation (5), the RMD value of level II can be calculated to be 0.7. Similarly, for level I, the VFS model parameters (c, a, M, b, d) are 0.1, 0.1, 0.1, 0.3, and 0.5, respectively, and $x_1$ is between b (0.3) and d (0.5). The RMD of level I can be calculated as 0.4 by introducing the above parameters into Equation (8). According to the characteristics described in Equation (26), the RMD is 0.1 for level III and is 0 for level IV.

For $\sigma_c/\sigma_t$, the eigenvalue ($x_2 = 23.97$) is located in the interval of level III. The VFS model parameters (c, a, M, b, d) of level III are 40, 26.7, 20.6, 14.5, and 1.57, respectively. Based on this, the eigenvalue $x_2$ belongs to the interval $[a, M]$. Inserting the above parameters into Equation (5), the RMD for level III is 0.724. Similarly, the VFS model parameters (c, a, M, b, d) of level IV are 26.7, 14.7, 1.5, 1.5, and 1.57, respectively. Therefore, $x_2$ is between c and a and the RMD for level IV can be calculated by introducing the above parameters into Equation (6) with a value of 0.112. The RMDs of level II and level I are 0.388 and 0, respectively, based on characteristic formula of Equation (26).

As for $w_{et}$, the eigenvalue ($x_3 = 6.6$) is greater than the pseudo-boundary value 6.5 of level IV. Equation (29) shows that RMD is equal to 1 for level IV and to 0 for other levels.

Therefore, the RMD matrix of these three indicators in Tianshengqiao II hydropower station can be expressed as follows:

$$u_{ij}(x) = \begin{bmatrix} 0.400 & 0.700 & 0.100 & 0.000 \\ 0.000 & 0.388 & 0.724 & 0.112 \\ 0.000 & 0.000 & 0.000 & 1.000 \end{bmatrix} \tag{41}$$

Similarly, the RMD matrix of all rockburst cases can be obtained by repeating the above process.

(3) The Initialized SRMD

The comprehensive evaluation model of Equation (10) can transform RMD into SRMD, but the weight of each index and model parameters $\alpha$ and $p$ should be determined first. In this paper, $\alpha = 2$ and $p = 1$ fuzzy optimization models are selected and eight different weight assignments are obtained from the references in Table 1. Based on the above analysis, combined with RMDs and eight weight assignment values, the initial SRMD can be calculated through Equation (10). Then, the initial SRMD is introduced into Equation (17) to obtain the level eigenvalue H, as shown in Figure 10a.

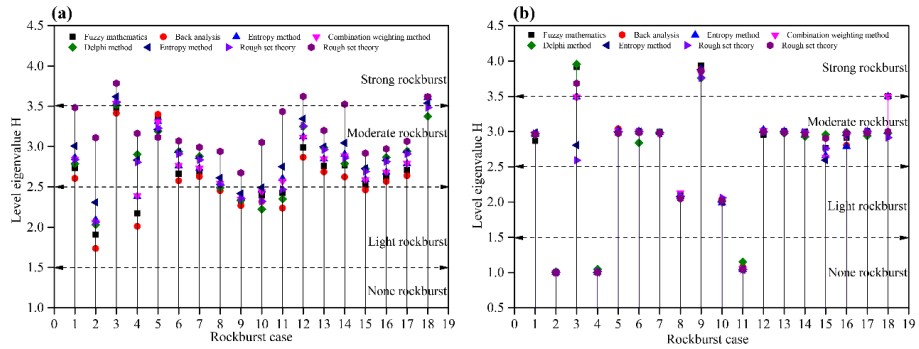

**Figure 10.** The prediction results of 18 rock burst cases before and after SRMD optimization: (**a**) initial SRMD and (**b**) optimized SRMD under eight different weight assignment conditions.

(4) The Optimized SRMD

In order to optimize the SRMD, a three-layer BP neural network is established. The nodes of the input layer, the hidden layer, and the output layer are 4, 5, and 4, respectively. The input of the network is the initial SRMD value, and the expected output O corresponds to the actual rockburst strength level of each rockburst case. In order to facilitate classification and calculation, the four strength levels are converted into the following forms:

$$O = \begin{bmatrix} Level\ 1 \\ Level\ 2 \\ Level\ 3 \\ Level\ 4 \end{bmatrix} = \begin{bmatrix} o_1 \\ o_2 \\ o_3 \\ o_4 \end{bmatrix} = \begin{bmatrix} 1 & 0 & 0 & 0 \\ 0 & 1 & 0 & 0 \\ 0 & 0 & 1 & 0 \\ 0 & 0 & 0 & 1 \end{bmatrix} \tag{42}$$

The initial SRMD values of the first 14 cases are selected as training samples of the three-layer BP neural network. Combining the expected output O and the weight adjustment models of Equation (36) and (37), the connection weight $w_{jk}$ between the input layer and the hidden layer and the connection weight $w_{kh}$ between the hidden layer and the output layer can be calculated (where the learning rate $\gamma$ is 0.05 and the network error E is 0.03). Then, based on the trained connection weight, the initial SRMD of the last four rockburst cases are input into the network as test samples for training and the network output is the optimized SRMD values of each rockburst case, of which the values are brought into Equation (17) to calculate the level eigenvalue H. The results are shown in Figure 10b.

*4.3. Discussion*

From Figure 10b, it can be seen that the predicted results of 18 rockburst cases based on the improved VFS method are basically consistent with the actual rockburst strength grade, except for No. 3, 5, and 9. No. 3, 5, and 9 belong to mixed-grade rockburst, and the actual rockburst grades are III~IV, II~III, and III~IV, respectively.

The new method improves the traditional VFS method from two aspects: simplifying the RMD calculation process and optimizing the SRMD. Then, what are the advantages of the new method compared with the traditional VFS method after such improvement? The following are discussed in detail:

(1) The improved VFS method has higher computational efficiency: Equation (41) can clearly show that the simplified RMD calculation method can reduce the number of RMD function operations at least two times compared with the traditional RMD calculation method. With the number increase of classifications level, indicators, and samples, this advantage will become more prominent. Since the distribution characteristics of RMD functions at different levels are known in advance (Figures 3c, 4c and 5c), the simplified RMD calculation method is simpler and more direct for RMD calculation, which greatly improves the operation efficiency of the improved VFS method.

(2) The improved VFS method can verify the correctness of RMD calculation results at all times: RMD is the core theme of the improved VFS method, which concerns whether the final prediction results are correct or not. Therefore, the effective guarantee of the correctness of RMD calculation results is the premise of obtaining high-precision prediction results. According to RMD relationship characteristic Equations (20), (26), and (28), the range of traditional RMD value [0, 1] is reduced. For example, the eigenvalue of $\sigma_\theta / \sigma_c$ in Tianshengqiao II hydropower station is located in the interval of level II, so $0.5 \leq u_{12}(x) \leq 1$, $0 \leq u_{11}(x) \leq 0.5$, and $0 \leq u_{13}(x) \leq 0.5$. In addition, $u_{11}(x) + u_{13}(x) = 0.5$. These features can be used to verify the correctness of RMD calculation results.

(3) The improved VFS method has higher prediction accuracy: From Figure 10a, it can be found that the prediction results based on the traditional VFS method are not ideal regardless of the weight. For example, the predicted results of No. 2, 4, and 11 are totally inconsistent with the actual strength grade. However, the prediction accuracy is significantly improved after the optimization of the initial SRMD using the BP neural network. This phenomenon shows that the combination of BP neural network and the VFS method can improve the classification and prediction ability of the model.

(4) The improved VFS method has higher fault tolerance and practicability: From Figure 10a, we can also find that the prediction results of traditional VFS method are easily affected by the weight values and that different weight values may lead to different prediction results, resulting in misjudgment. For example, the prediction results of No. 2–4, 8–12, 14, and 18 rockburst cases span different levels in different weights. However, after the initial SRMDs are optimized by using the fuzzy BP optimization neural network model, the prediction results are basically consistent under different weights, which is within the same strength level, except for No. 3. The above results show that the improved VFS method has higher fault tolerance and anti-jamming ability and that its dependence on weight is low, so it has better practicability.

## 5. Conclusions

From the above analysis, we can draw the following conclusions.

(1) This research improved the traditional VFS method from two aspects: (i) simplifying the RMD calculation process and (ii) optimizing SRMD, which was applied to the prediction of rockburst strength. Good results have been achieved.

(2) Compared to the traditional VFS method, the improved VFS has a clearer, more efficient calculation process and a more credible and stable prediction. The improved VFS method simplified the RMD



calculation process by using the characteristic relationship of RMD at different levels and verified the correctness of RMD calculation results through these characteristics at all times. Besides, the improved VFS also uses the BP neural network to optimize the SRMD, which improves the prediction accuracy of SRMD. By this way, the influence of weight change on SRMD is also effectively avoided. Therefore, the improved method has higher fault tolerance rate and anti-jamming ability.

(3) The original index data, the rockburst occurrence, and the intensity in underground projects all have certain dynamic variability and fuzziness, so it is difficult to express the rockburst criterion with an accurate relational expression. A more reasonable results can be obtained by using RMD and SRMD to predict. However, the application of improved VFS to rockburst prediction is still in the phase of theory and there are still some problems to be further explored, for example, how to reasonably construct RMD function, how to select and calculate the layers of BP neural network, the number of nodes and the connection weight matrix, etc. to make the rockburst prediction results more in line with the actual situation, especially for the accurate prediction of mix-level and intermediate-level rockburst.

**Author Contributions:** Formal analysis, Y.X.; Funding acquisition, L.N. and Y.X.; Investigation, T.Z. and Y.W.; Methodology, H.W.; Validation, Y.L. and C.D.; Visualization, Y.H.; Writing—original draft, H.W.

**Funding:** This work was supported by the National Natural Science Foundation of China (Grant No. 41702300 and Grant No. 41572254).

**Conflicts of Interest:** The authors declare no conflict interest.

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
