# Peer review of "Comprehensive Prediction and Discriminant Model for Rockburst Intensity Based on Improved Variable Fuzzy Sets Approach"

_applsci, doi:10.3390/app9153173_

Round 1

Reviewer 1 Report

The article contains very important contents related to the rock burst risk prediction. Underground exploitation of mineral resources is inseparably connected with natural hazards, from which rock bursts are still subject to research and simulation. The article is well written, some of the suggestions below are intended to better understand the issue and phenomena occurring in rock mass. In the introduction, please note that in conditions of tremors, one of the lines of defense against this threat is a special support capable of absorbing energy. Two publications should be added regarding this issue, namely: https://doi.org/10.1051/e3sconf/20187100006 and https://doi.org/10.1515/sgem-2017-0029 In point 2 in table 1, please write which program is used for back analysis. In point 3 in figures 3c and 5c, please explain how the value was calculated at the level above 0.5 In point 4 in table 2, please explain what decided the choice of these three main prediction criteria. Ultimately, I think that the article is scientific in nature and is related to natural hazards associated with underground exploitation.

Reviewer 2 Report

Dear Authors,

The paper Comprehensive prediction and discriminant model for rockburst intensity based on improved variable fuzzy sets approach by Hong Wang, Lei Nie, Yan Xu, Yan Lv, Yuanyuan He, Chao Du, Tao Zhang, Yuzheng Wang  is well suited for journal Applied Sciences. The authors of this article analyzed present the improved VFS method for rockburst intensity prediction.

The paper is interesting and scientifically valuable. The paper contains parts in good order: introduction, description of the VFS method, description of the improved VFS method, results on the example and discussion, conclusions.

The article was written enough well in English, is understandable for a reviewer, a person who does not speak English as a mother tongue.

For the reasons stated, I support publication of the paper in the journal.

Major shortcomings:

- Not found.

Minor shortcomings:

- In 3 places the text has been moved to the new line too early – line: 100, 318, 432
